# Detecting papilloedema as a marker of raised intracranial pressure using artificial intelligence: A systematic review

Lekaashree Rambabu[1,2,3,4‡]*, Thomas Edmiston[5,6‡], Brandon G. Smith[2,6],
Katharina Kohler[2,4,6], Angelos G. Kolias[2,7], Richard A. I. Bethlehem[8], Pearse A. Keane[9,10],
Hani J. Marcus[11,12], EyeVu Consortium[2], Peter J. Hutchinson[2,7], Tom Bashford[2,4,6,13]

1 Department of Medicine, University of Cambridge, Cambridge, United Kingdom, 2 NIHR Global Health Research Group on Acquired Brain and Spine Injury, University of Cambridge, Cambridge, United Kingdom, 3 Accelerate Programme for Scientific Discovery, Department of Computer Science and Technology, University of Cambridge, Cambridge, United Kingdom, 4 Department of Anaesthesia, Addenbrooke's Hospital, Cambridge, United Kingdom, 5 School of Clinical Medicine, University of Cambridge, Cambridge, United Kingdom, 6 International Health Systems Group, Department of Engineering, University of Cambridge, Cambridge, United Kingdom, 7 Division of Academic Neurosurgery, Addenbrooke's Hospital, Cambridge, United Kingdom, 8 Department of Psychology, University of Cambridge, Cambridge, United Kingdom, 9 Institute of Ophthalmology, University College London, London, United Kingdom, 10 NIHR Biomedical Research Centre, Moorfields Eye Hospital and UCL institute of Ophthalmology, London, United Kingdom, 11 Wellcome/EPSRC Centre for Interventional and Surgical Sciences, University College London, London, United Kingdom, 12 Victor Horsely Department of Neurosurgery, National Hospital for Neurology and Neurosurgery, London, United Kingdom, 13 Cambridge Public Health Interdisciplinary Research Centre, University of Cambridge, Cambridge, United Kingdom

¶ The membership of the EyeVu Consortium is provided in the acknowledgements.
‡ denotes joint first authorship.
* lr593@cam.ac.uk

## Abstract

Automated detection of papilloedema using artificial intelligence (AI) and retinal images acquired through an ophthalmoscope for triage of patients with potential intracranial pathology could prove to be beneficial, particularly in resource-limited settings where access to neuroimaging may be limited. However, a comprehensive overview of the current literature on this field is lacking. We conducted a systematic review on the use of AI for papilloedema detection by searching four databases: Ovid MEDLINE, Embase, Web of Science, and IEEE Xplore. Included studies were assessed for quality of reporting using the Checklist for AI in Medical Imaging and appraised using a novel 5-domain rubric, 'SMART', for the presence of bias. For a subset of studies, we also assessed the diagnostic test accuracy using the 'Metadta' command on Stata. Nineteen deep learning systems and eight non-deep learning systems were included. The median number of images of normal optic discs used in the training set was 2509 (IQR 580–9156) and in the testing set was 569 (IQR 119–1378). The number of papilloedema images in the training and testing sets was lower with a median of 1292 (IQR 201–2882) in training set and 201 (IQR 57–388) in the testing set. Age and gender were the two most frequently reported demographic data, included by

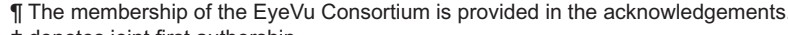

**Data availability statement:** All data are in the manuscript and/or supporting information files.

**Funding:** This research was funded by the NIHR (Grant Number - NIHR132455; URL - https://fundingawards.nihr.ac.uk/award/NIHR132455) using UK international development funding from the UK Government to support global health research. The views expressed in this publication are those of the author(s) and not necessarily those of the NIHR or the UK government. PJH is supported by the NIHR Cambridge Biomedical Research Centre, the NIHR Brain Injury MedTech CoOperative. BGS, AK, TB, and PJH are supported by the NIHR Global Health Research Group on Acquired Brain and Spine Injury. PJH is also supported by NIHR (Senior Investigator award, Cambridge BRC, Healthtech Research) and the Royal College of Surgeons of England. LR is supported by the NIHR Academic Clinical Fellowship. HJM is funded by the Wellcome/EPSRC Centre for Interventional and Surgical Sciences (203145/Z/16/Z) and the NIHR UCLH/UCL Biomedical Research Centre. PAK is supported by a UK Research & Innovation Future Leaders Fellowship (MR/T019050/1). KK is supported by a NIHR Development and Skills Enhancement Grant. The funders had no role in study design, data collection and analysis, decision to publish, or preparation of the manuscript.

**Competing interests:** PK has acted as a consultant for Retina Consultants of America, Topcon, Roche, Boehringer-Ingleheim, and Bitfount and is an equity owner in Big Picture Medical. He has received speaker fees from Zeiss, Novartis, Gyroscope, Boehringer-Ingleheim, Apellis, Roche, Abbvie, Topcon, and Hakim Group. He has received travel support from Bayer, Topcon, and Roche. He has attended advisory boards for Topcon, Bayer, Boehringer-Ingleheim, RetinAI, and Novartis. RB holds shares and is a director of centile Bioscience Inc. LR, TE, BGS, KK, AGK, HJM, PJH, and TB don't have any competing interests to declare.

one-third of the studies. Only ten studies performed external validation. The pooled sensitivity and specificity were calculated to be 0.87 [95% CI 0.76-0.93] and 0.90 [95% CI 0.74-0.97], respectively. Though AI model performance values are reported to be high, these results need to be interpreted with caution due highly biased data selection, poor quality of reporting, and limited evidence of reproducibility. Deep learning models show promise in retinal image analysis of papilloedema, however, external validation using large, diverse datasets in a variety of clinical settings is required before it can be considered a tool for triage of intracranial pathologies in resource-limited areas.

---

## Author summary

Papilloedema is a condition characterised by the swelling of the optic disc in the eye. It can be caused by increased intracranial pressure. It can be caused by an increase in pressure within the cranial cavity, which may be due to traumatic brain injuries, tumours, or infections. In low-resource settings where access to specialist imaging is limited, identifying papilloedema with accuracy by the bedside using retinal images and artificial intelligence could potentially serve as a tool for triaging when raised intracranial pressure is suspected and urgent surgical or medical intervention may be required. Our systematic review has critically appraised the primary studies which use any form of artificial intelligence to detect papilloedema from images of the retina. We provide a comprehensive overview and an in-depth discussion on the quality of reporting, areas of bias in model design, common limitations, and key findings from the primary literature.

## 1. Introduction

Papilloedema is swelling of the optic disc secondary to raised intracranial pressure (ICP), which may be a result of cerebral haemorrhage, space occupying lesions, trauma, infections of the central nervous system, idiopathic intracranial hypertension (IIH), and disorders impairing cerebral venous drainage [1,2]. The severity of papilloedema is commonly graded using the Frisén scale, from stage 0 (normal optic disc) to 5 (severe) (Fig 1) [3]. Over time, raised ICP causes an increase in cerebrospinal fluid (CSF) pressure around the optic nerves, resulting in intraneuronal ischaemia and visual loss [1,4]. In addition, raised ICP is itself a life-threatening emergency which can lead to permanent neurological deficits or death, and requires urgent investigation and management.

Clinically, papilloedema may present with headaches, nausea and blurred vision, and in the early stages the most common visual field change is an enlargement of the blind spot [5]. Other possible symptoms include horizontal diplopia due to an abducens nerve palsy, pulsatile tinnitus, or focal neurological deficits related directly to the elevated ICP [4,6]. Suspected papilloedema should be investigated with both a

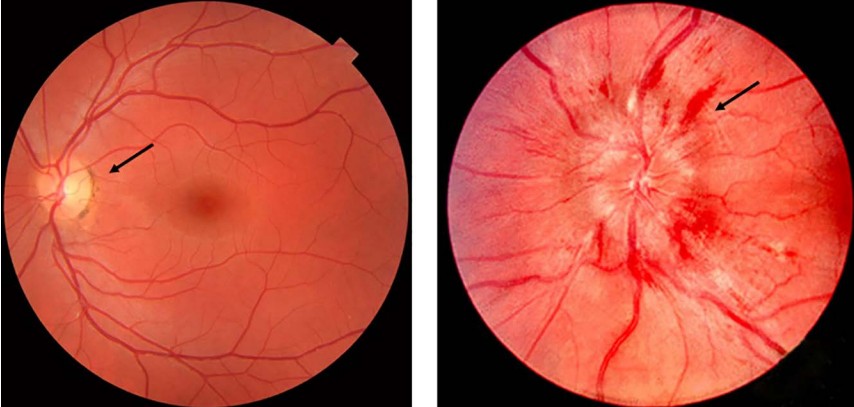

**Fig 1. Fundus photographs of normal optic disc and papilloedema.**

neurological and ophthalmic examination, as well as appropriate imaging. Management depends on the underlying cause and any associated complications. This may include surgical intervention, pharmacological treatments such as diuretics or steroids, or conservative approaches such as weight loss or low-sodium diet, all aimed at reducing complications from raised ICP and minimising visual loss [1].

The reported incidence of papilloedema in the literature is variable, influenced by factors such as the underlying aetiology, the location and extent of the intracranial pathology, the duration of raised intracranial pressure, the clinical setting in which the papilloedema is identified, and the year of the published study – particularly whether it was conducted prior to when neuroimaging was more accessible and when certain conditions such as brain tumours were diagnosed at more advanced stages [1,7–10]. Incidence rates have ranged from 28% to 80% in patients with brain tumours, 10% to 24% in patients with aneurysmal subarachnoid haemorrhage, and 3.5% in patients with traumatic brain injuries (TBI) [1]. Papilloedema can also be absent despite the presence of raised ICP in acute conditions such as spontaneous haemorrhage or trauma [10], and may not always correlate directly with the degree of raised ICP [1]. However, the presence of papilloedema in TBI has been associated with poorer outcomes [11,12]. In an observational study by Mattar et al. in 2020 involving 72 patients with moderate TBI (defined as those with an admission GCS between 9–12), 51.3% of patients had papilloedema and those with early and persistent papilledema had the worst functional outcomes [12]. Only 3 out of 23 (13%) cases of early papilloedema patients had the highest Glasgow Outcome Score of 5 (suggesting 'good recovery'), compared to 7 out of 14 (50%) with those who had papilloedema on day 3 and 18 out of 35 (51.4%) who didn't have any papilloedema at all. These findings were supported by another recent observational study by Meena et al. in 2025 involving 120 patients with moderate brain injury [13]. They demonstrated that patients with papilloedema had significantly more contusions than those who did not on CT scans [13]. Moreover, the absence of papilloedema showed a 70.8% predictability for 'good recovery' when measured by the Glasgow Outcome Scale 72 hours after admission [13]. These results show the value of papilloedema as an important prognostic factor in traumatic brain injuries, suggesting that early identification has the potential to lead to reduced morbidity and mortality. It has been estimated that 50% of the global population will suffer a TBI at some point in their life, with prevalence believed to be three times higher in low- and middle-income countries (LMICs) than high-income countries (HICs) [14]. In resource-limited areas, access to rapid neuroimaging may be lacking due to cost, geographical distance, or due to limited numbers of radiologists for interpretation [15]. It has been speculated that the ability to accurately detect papilloedema in these settings may be a useful bedside assessment for triage and transfer to specialist services. However, there is little literature describing the prevalence and aetiology of papilloedema in resource-poor countries although differences to high-income populations may be hypothesised to exist due

to genetic, developmental, and acquired characteristics. As an example, African American patients have been shown to present with more aggressive ocular disease and suffer worse visual outcomes in IIH compared to Caucasian Americans, and require a lower threshold for intervention as well as closer follow-up [16,17].

Diagnosis of papilloedema is challenged by several factors, most notably the experience in ophthalmoscopy required [18–20], the presence of conditions that mimic papilloedema (e.g., optic nerve head drusen, congenital abnormalities, tumours and hamartomas) [21], and the need for pharmacological dilatation of the pupil [20]. In well-resourced hospitals, these may be overcome through access to specialist ophthalmologists, provision for alternative ophthalmological testing (e.g., optical coherence tomography, ultrasound or intravenous fluorescein angiography), and a greater availability of neuroimaging. However, in resource-poor environments, the diagnosis of papilloedema remains a challenge.

Smartphone-based ophthalmoscopy driven by artificial intelligence (AI) algorithms for automated interpretation could be a promising candidate as an assisted diagnostic tool, particularly in resource-poor settings where access to specialist services may not be readily available. However, this remains a relatively undeveloped area of research and is complicated by the lack of high-quality image datasets from resource-poor populations.

The aim of this systematic review was to provide a comprehensive overview of the current state of evidence regarding the use of AI in ophthalmic imaging for the diagnosis of papilloedema. To meet this aim, we set out to describe and critique the published literature, and to answer the following research question(s):

1. What is the current state of research on the use of AI in ophthalmoscopy for papilloedema screening?

2. What algorithms or neural network architectures have been demonstrated to be efficacious in identifying papilloedema from ophthalmoscopy images?

3. What datasets, if any, have been used to train the models and are these datasets open-source and readily available?

## 2. Materials and methods

The reporting of this study is in accordance with the Preferred Reporting Items for Systematic Reviews and Meta-analyses (PRISMA) 2020 guidelines [22]. The study was prospectively registered with PROSPERO (CRD42023413049). A full protocol was submitted for peer-review and published online in December 2023 (https://doi.org/10.1097/SP9.0000000000000016).

### 2.1 Search strategy

The final search strategy was developed to identify original studies reporting the use of computational algorithms to identify papilloedema from retinal images taken using an ophthalmoscope. The search strategy was executed on the 21st April 2023 and repeated again on 11th October 2024 across four key databases: Ovid MEDLINE, Ovid Embase, Clarivate Web of Science and IEEE Xplore, retrieving articles from database inception. An example strategy for Ovid MEDLINE is outlined in S1 Appendix, along with definitions of AI and its subsets referred to in the paper.

### 2.2 Selection of studies

Studies were selected in two stages. First, all titles and abstracts were independently screened by two reviewers (BGS and LR) against predefined inclusion (PCC) criteria (S2 Appendix) using the web-based review collaboration tool Rayyan (Fig 2) [23]. Potentially eligible studies were then retrieved for full-text review. Conflicts at each stage were resolved by discussion and consensus with a third reviewer (TB).

### 2.3 Data extraction, charting, and critical appraisal of reports of AI with clinical applications

Data extraction was conducted by three reviewers (LR, BGS, and TE) and tabulated in MS-Excel for critical appraisal. We used the Checklist for Artificial Intelligence in Medical Imaging (CLAIM) for assessing reporting quality and a new

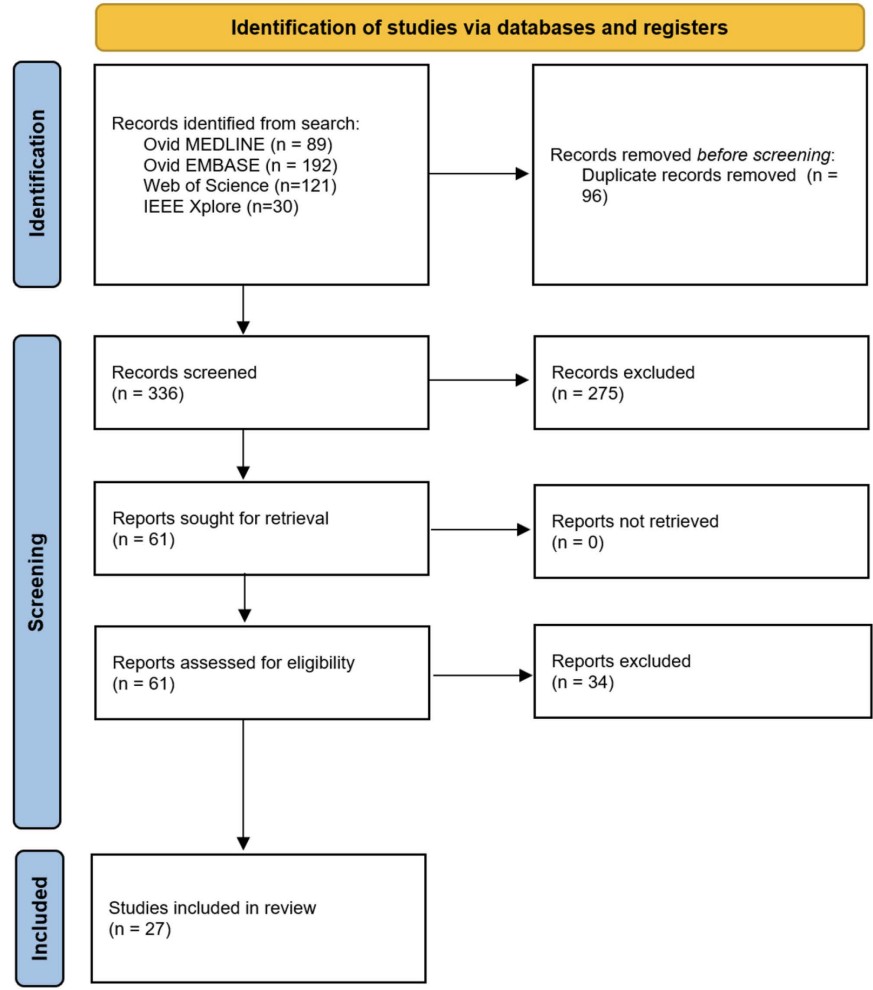

**Fig 2. PRISMA flow diagram of included and excluded studies.**

framework, SMART. SMART comprises five domains: (1) Study design, (2) Management and Handling of data, (3) Algorithm, modes and performance, (4) Real-life clinical implications of the model, and (5) Transparency, accountability, medicolegal and ethical considerations (Fig 3). SMART was adapted based on previous work and existing tools for assessing diagnostic and prognostic predictive machine learning studies (TRIPOD, CHARMS, and ROBUST-ML) [24–26]. Each domain was refined through iterative discussion among the study authors LR and BGS with consensus endorsed by the broader authorship team with expertise in machine learning, data science, and medical imaging. While we did not assign formal weights to each domain, we treated them as equally important for the purpose of critical appraisal, acknowledging the different roles they play in evaluating AI models.

Though there are several frameworks such as CONSORT-AI, DECIDE-AI, STARD-AI, SPIRIT-AI and TRIPOD-AI, which exist to assess reporting quality of AI studies, we chose CLAIM as it is specifically tailored to assessing quality in medical image analysis studies that use AI. CLAIM is a checklist developed by Mongan et al. in 2020 and comprises 42 items that serve as a 'best practice' guide for authors [27]. We evaluated the reporting of the studies against this checklist, and assigned a score for each item depending on the quality of reporting as follows: 0 for not reported, 1 for poorly reported (<50% of the item criteria reported), 2 for partially reported (50–99% of the item criteria reported) and 3 for fully

## 'SMART' critical appraisal framework for Artificial Intelligence studies

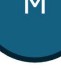

**Study design**
Examine differences between studies and presence of bias in:
- Machine learning method: type, frameworks, packages or libraries used
- Datasets: Source, accessibility, sample size for training, validation and testing, use of sample size calculation
- Participant characteristics: eligibility criteria, discussion of patient demographics and diversity within dataset
- Ground truth: definition, presence of any ambiguity, use of objective methods to minimise bias

**Management and Handling of data**
Examine differences between studies and presence of bias in:
- Storage practice of dataset upon acquisition: data organisation and location, level of data security, and accessibility
- Pre-processing practices and rationale for chosen practices
- Practices of handling missing data, outliers or any other inconsistencies
- Data augmentation techniques applied to the dataset and provided reasoning for chosen technique
- Handling of class imbalance between different categories in a dataset

**Algorithms, models, and performance**
Compare and contrast:
- Study and participant characteristics within studies
- Performance metrics for models used (e.g. precision, recall, accuracy, F1 score, sensitivity, and Area Under Receiving Operator Characteristics Curve) and rationale for chosen metric
- Methods used to improve model interpretability and transparency

**Real-life clinical implications of the model**
Consider if:
- Authors have provided a description of model's possible integration with existing workflows
- Authors described the efficacy of the model in improving patient care and/or outcomes
- Authors, where relevant, detail the purpose of the model, its clinical need, and transferability to clinical practice

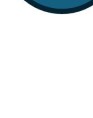

**Transparency, accountability, medicolegal and ethical considerations**
Consider if authors have explicitly discussed:
- The presence of algorithmic bias and fairness in their model
- Transparency of their datasets, algorithm code, and methods used to assess performance
- Model's potential to cause harm
- Wider ethical, legal, and social concerns

**Fig 3. SMART – framework for critical appraisal of clinical applications of AI.**

reported. We then calculated the total study score and converted it to a percentage of the maximum possible score for each section, and for the total score overall, to enable a quantitative comparison of reporting quality. An exception was made for Vasseneix et al. 2023, where much of the methodology was provided in a referenced study, so the relevant criteria were excluded from score calculations [28]. We also calculated the average score across the studies for each item, to determine which items were well reported and which were poorly reported which can be viewed in S3 Appendix.

## 2.4 Data synthesis and analysis

Two reviewers (LR and TE) cross-checked data from the studies to determine and extract relevant parameters. These parameters were grouped into: participant demographics; data source and characteristics; model training and validation; and model performance. These are summarised in Table 1.

Where these parameters were not included in a paper, this was defined as 'not reported (NR)'. The studies were subdivided by type of machine learning algorithm used, and descriptive statistics were applied to summarise the number of participants, age, gender, normal optic disc images, and papilloedema images used to train and test the model respectively. To perform a meta-analysis of the model performance, we constructed 2x2 contingency tables using data from studies that fully reported the number of images used in testing data, as well as the sensitivity and specificity of their model. Where studies reported performing external validation, we used these data; however, not all studies explicitly stated whether their testing set was internally or externally validated. We used the 'Metadta' Stata command to perform a meta-regression of the data [29]. This model assesses both single-study variability, using binomial distributions, and multi-study variability or heterogeneity. The model then presents the results as a forest plot and a summary receiver operating characteristic (SROC). The SROC contains a single summary operating point, which is the average sensitivity and specificity of the studies, with a 95% confidence region surrounding the summary point. An SROC is an effective way to assess the combined diagnostic test validity of the different studies, providing a good picture of the overall state-of-the-art and the suitability of the existing models for implementation into clinical practice.

## 3. Results

Following deduplication, a total of 336 unique articles were identified for screening, with 27 included in the final dataset. The number of articles retrieved by the search strategy, screened, and included or excluded at title and abstract or full-text review (with accompanying reasoning) is reported in Fig 2. We included twenty-seven studies conducted between 2011 and year 2024, of which nineteen used deep learning systems, five used support vector machines, one used a linear mixed model, and two used a random forest classifier [28,30–55]. First, we summarise the reporting quality assessment of the studies in this section. Following this, we provide our critical appraisal of the included studies using the SMART framework.

The assessment of reporting quality revealed that most papers did not meet the reporting standards outlined by CLAIM (Fig 4) [27]. The introduction was the only section where the majority of studies (n = 26) met more than 50% of CLAIM's criteria. The results section was the least well reported, with clinical and demographic information missing from most of the studies. Ethnicity data was formally collected by only three studies [32,53,54], though it was not available for all images included in the study and the split between the different ethnicities was not reported in one [32]. Four studies

**Table 1. Summary of extracted parameters.**

| Participant demographics | Data source and characteristics | Model training and validation | Model performance |
|---|---|---|---|
| Number of participants in training and testing data | Source of data | Machine learning model and/or framework | Accuracy |
| Number of images of normal retina in testing and training data | Exclusion criteria for poor quality images (if used) | Reference standard | Sensitivity |
| Number of images of papilloedema in training and testing sets | Image quality | Internal validation performed? | Specificity |
| Age | Model of camera used | External validation performed? | AUROC |
| Gender | Accessibility of data (is it open access?) | Transfer learning applied? | |
| City/country of origin of participants | | | |
| Ethnicity data | | | |

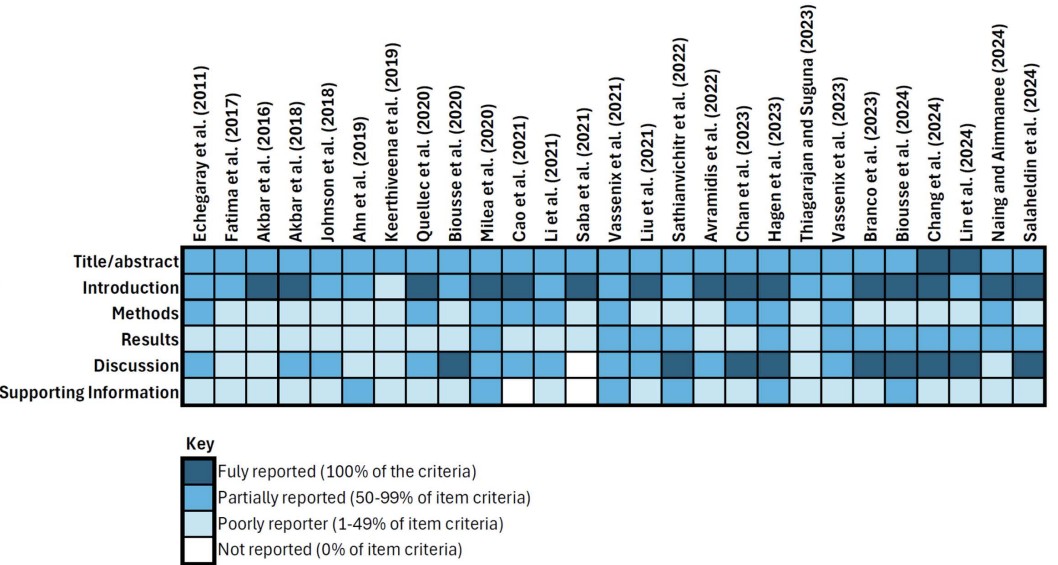

**Fig 4. Reporting quality assessment of included studies using CLAIM.**

partially report the ethnic background of their study cohort [32,39,49,51].Though model performance was reported by all the studies, benchmarking the performance of the AI model against current standards such as by comparing its performance with medical experts was rarely performed. A detailed analysis of the reporting quality is available in S3 Appendix.

### 3.1 Study design of DLS and non-DLS models

A Deep Learning System (DLS) is a subset of AI that uses multi-layered neural networks for tasks that would typically require human intelligence such as making predictions, classifying abnormal and normal pathology, identifying relationships between different variables, or making decisions [56]. Non-DLS models which use supervised machine learning (ML) techniques offer the advantage of being relatively more interpretable than deep learning systems and are particularly well suited for tasks that are dependent on dimensionality reduction such as medical image analysis. Non-DLS models which use supervised machine learning (ML) techniques offer the advantage of being relatively more interpretable than deep learning systems and are particularly well suited for tasks that are dependent on dimensionality reduction such as medical image analysis. We included nineteen DLS studies with key features as summarised in Table 2.

**3.1.1 Summary of architectures used and model purpose.** All the DLS studies that were included had a convolutional neural network (CNN) as their main architecture, although the architecture used and the layer modification differed depending on the task required and the dataset used. Commonly used CNN architectures were: Inception-V3, Inception-V4, ResNet101, DenseNet, SeResNe50, U-Net, GoogleNet, EfficientNet, and VGGNet, most of which were reported to be pre-trained on ImageNet datasets and modified based on model needs, with the exception of three studies [40,47,50]. Four out of five non-DLS studies that utilised support vector machines performed optic disc detection as a region of interest as part of the pre-processing step, followed by feature extraction of relevant features, and subsequent classification with SVM between normal and papilloedema images.

**3.1.2 Dataset acquisition.** Datasets were retrospectively obtained by a majority of the studies, with the exception of one mixed linear model study that prospectively collected data from two Danish regional tertiary hospitals that specialised with idiopathic intracranial hypertension [46]. Publicly available datasets for papilloedema are scarce as reported in

**Table 2. Summary of all studies' design, datasets used, and image quality.**

| Author, year | Location | Type of algorithm | Camera used | Image quality | Dataset | Publicly available? | Papilloedema ground truth | Internal validation | External validation |
|---|---|---|---|---|---|---|---|---|---|
| Biousse et al., 2024 | USA | Deep Learning System | Kowa α-D, Torrence, California, USA | NR | FOTO-ED | No | Neuro-ophthalmologist classification | N/A | Yes |
| Chang et al., 2024 | USA | Deep Learning System | NR | NR | Retrospective multi-centre study | No | Neuro-ophthalmologist classification | Yes | No |
| Branco et al., 2024 | USA | Deep Learning System | NR | NR | IIHTT Trial and Images from University of Iowa | No | Neuro-ophthalmologist classification | Yes | No |
| Naing and Aimmanee, 2024 | International | Supervised ML with SVM | NR | Variable - 600x600; 2300x 1900; 2144x 1424; 4288 x 2848; 512x512; 2048x1536 | RFmiD datasets, Hoyt and Cent | Partially | NR | Yes | NR |
| Lin et al., 2024 | International | Deep Learning System | NR | NR | Retrospective multi-centre study with data from 3 centers in Atlanta, USA, Bucharest-Romania, and Singapore | No | Neuro-ophthalmologist classification | N/A | Yes |
| Salaheldin et al., 2023 | Egypt | Deep Learning System | NR | 224x224x3 | South Valley University | No | NR | Yes | Yes |
| Vasseneix et al., 2023 | International | Deep Learning System | Topcon TRC 50DX and TRC NW8, Topcon Medical-Systems, Oakland, NJ; Kowa Nonmyd 7 and VX-10, Kowa Company, Tokyo, Japan | NR | BONSAI | No | Neuro-ophthalmologist classification | Yes | Yes |
| Thiagarajan and Suguna, 2023 | India | Deep Learning System | NR | NR | Kaggle Ungsoo Kim's dataset | Yes | NR | Yes | No |
| Liu et al., 2023 | International | Deep Learning System | NR | Partially reported. ~2056 x 2048 pixels for 86% of abnormal images in testing dataset. DRIONS database contained images of 600 x 400 resolution | NRM practice, & ACRIMA for training. PSS practice, Paxos, & DRIONS for testing | Partially | Partially reported – for NRM described as 'combination of detailed clinical history, clinical examination, ophthalmic imaging, and/or neurological imaging' | Yes | Yes |
| Chan et al., 2023 | International | Deep Learning System | Topcon TRC 50DX and TRC NW8, Topcon Medical Systems, Oakland, NJ; Kowa Nonmyd 7 and VX-10, Kowa Company, Tokyo, Japan | 456 x 456 | BONSAI | No | Ophthalmologists | Yes | Yes |

**Table 2.** (Continued)

| Author, year | Location | Type of algorithm | Camera used | Image quality | Dataset | Publicly available? | Papilloedema ground truth | Internal validation | External validation |
|---|---|---|---|---|---|---|---|---|---|
| Hagen et al., 2023 | Denmark | Linear Mixed Model | EpiCamM | 1280 x 1024 | Patients at two Danish tertiary hospitals | No | Lumbar opening pressure | Yes | No |
| Sathianvichitr et al. 2022 | International | Deep Learning System | NR | NR | BONSAI consortium, ODDS consortium, University of Calgary, University of Copenhagen, Western University, Ramathibodi Hospital, Mahidol University, Tehran University of Medical Sciences, Angers University Hospital and Farabi Eye Hospital | No | Ophthalmologists | Yes | Yes |
| Li et al., 2022 | China | Deep Learning System | NR | 512 x 512 | Henan Provincial Peoples' Hospital, Beijing Tongren Hospital, Beijing Aier Intech Eye Hospital, Peking Union Medical College Hospital | No | Ophthalmologist | Yes | Yes |
| Avramidis et al., 2022 | USA | Deep Learning System | NR | NR | Children's Hospital Los Angeles and University of California | No | Neuroimaging evidence of intracranial evidence of an intracranial lesion causing raised ICP or lumbar puncture with opening pressure >28 cm $H_2O$ | Yes | No |
| Saba et al., 2021 | International | Deep Learning System | NR | 700x605 | STARE | Yes | Expert annotations of STARE dataset | Yes | No |
| Vasseneix et al., 2021 | International | Deep Learning System | 15 different cameras used, including Canon/Zeiss/Topcon models | NR | BONSAI | No | Ophthalmologist classification | Yes | Yes |
| Quellec et al., 2020 | France | Deep Learning System | Canon CR-DGi or CR2, Topcon TRC-NW6 or TR-NW400 | 299 x 299 | OPHDIAT | No | Ground truth annotations obtained by combining structured information and manually-extracted textual information | Yes | Yes |
| Milea et al., 2020 | International | Deep Learning System | Canon CR-DGi or CR2, Topcon TRC-NW6 or TR-NW400 | 224 x 224 | BONSAI | No | Diagnoses provided by expert clinicians | Yes | Yes |

*(Continued)*

**Table 2.** (Continued)

| Author, year | Location | Type of algorithm | Camera used | Image quality | Dataset | Publicly available? | Papilloedema ground truth | Internal validation | External validation |
|---|---|---|---|---|---|---|---|---|---|
| Biousse et al., 2020 | International | Deep Learning System | Topcon TRC 50DX and TRC NW8, Topcon Medical Systems, Oakland, NJ; Kowa Nonmyd 7 and VX-10, Kowa Company, Tokyo, Japan | NR | BONSAI | No | Ophthalmologist classification | Yes | No |
| Keerthiveena et al., 2020 | International | Supervised ML with SVM | Topcon TRC-NW8 | 1844 x 1266 | DRIVE/STARE/PSGIMSR | Yes | Existing algorithm | Yes | No |
| Cao et al., 2020 | China | Deep Learning System | NR | 1500 x 1500 | Eye Centre at Second Affiliated Hospital of Zhejiang University School of Medicine | No | Ophthalmologist classification | Yes | No |
| Ahn et al., 2019 | South Korea | Deep Learning System | AFC-330, Nidel, Japan | 240 x 240 | Kim's Eye Hospital | Yes | NR | Yes | No |
| Akbar et al., 2017 | Pakistan | Supervised ML with SVM and RBF | Local dataset used TOPCON TRC-NW8 | VIVAVR - 768 x 584 pixels STARE - 605 x 700 pixels AVRDB - 1504 x 1000 pixels | INSPIRE-AVR, VICAVR, STARE and local dataset AVRDB | Partially | Not explicitly stated. Local images annotated through ophthalmologists. | Yes | No |
| Johnson et al., 2018 | USA | Random Forest Classifier | NR | NR | NR | NR | Optical coherence tomography images | Yes | No |
| Akbar et al., 2017 | Pakistan | Supervised ML with SVM and RBF | NR | NR | STARE and a local AFIO dataset | Partially | Not explicitly stated. Images annotated through 2 ophthalmologists. | Yes | No |
| Fatima et al., 2017 | Pakistan | Supervised ML with SVM | NR | 1504 x 1000 for local and 700 x 605 for STARE | STARE and a local AFIO dataset (states that only 10 images of papilloedema were present in STARE and 30 were acquired "online" without any source stated) | Partially | Not explicitly stated. Images annotated through ophthalmologists. | Yes | No |
| Echegaray et al., 2011 | USA | Supervised ML with decision tree forest | 2008 –2009; RC 50-DX retinal cameras [Topcon, Tokyo, Japan] with a Megavision 6 megapixel back | 2392 x 2048 | Local dataset | No | Papilloedema caused by raised ICP which was diagnosed via a LP; diagnosed by neuro-ophthalmologists | Yes | No |

Table 2. A detailed summary of the number of normal retinal images and papilloedema images reported are provided in Table A for DLS and Table B for non-DLS ML models in S5 Appendix. Neither of the two largest datasets of papilloedema images used are publicly accessible [30,32]. Only four DLS studies use datasets that are publicly available such as STARE, DRIONS, ACRIMA, and Kim Ungsoo's Kaggle dataset (Table 2) [33,38–40]. Classification performance of DLS depends on both the size and quality of the initial training datasets used, with performance generally improving with an increase in size of datasets [57]. Despite the emphasis on its importance in recent studies [58,59], estimating the required training dataset size was not performed by any of the DLS studies included in this review. Only one linear mixed model study estimated that they required a minimum of 16 participants based on their power analysis [46]. Though several sample-size estimating methods have been proposed, these are not commonly used due to the lack of standardised model-specific guidelines [57,59,60].

**3.1.3 Participant eligibility criteria.** Only six DLS and two non-DLS explicitly stated their inclusion and exclusion criteria [30,32,48,53–55]. However, the studies may vary in the level of detail they provide and diagnostic criteria they use. For instance, Li et al. (2022) included images that had a normal fundus and those with a selection of 12 fundus diseases, each defined by a specific diagnostic criterion [30]. Milea et al. (2020) reported comprehensive eligibility criteria in their supplementary file with the main inclusion criteria being fundus photographs of the optic disks and definitive corresponding clinical diagnoses made by expert neuro-ophthalmologists. The main exclusion criteria in both studies were insufficient image quality or a failure to annotate images with consistent labels. There is significant potential for bias in image classification studies due to uncertainty in diagnostic criteria, variation in methods for data collection, presence of duplicates, variable quality of images, and subsequent feature selection bias. In the absence of clear reporting of eligibility criteria in a majority of the studies included in the review, it is difficult to assess report risk of bias in participant or image selection.

**3.1.4 Reference standard and annotations.** The 'ground truth', also known as the 'reference standard', is 'the best available method for establishing the presence or absence of the target condition' [61]. The reference standard for twelve of the DLS studies was set by ophthalmologists, neuro-ophthalmologists, or 'expert clinicians' [30,32,40,41,43,44,47,49,51,53–55]. Only one study used neuroimaging or direct evidence (elevated lumbar puncture opening pressures) of raised intracranial pressure as a reference standard. Similar to DLS models, ophthalmologists or neuro-ophthalmologists were most commonly used for annotating and grading of all the papilloedema images based on the Friesen scale [34–37].

Three studies did not report reference standard setting [33,38,50], with one paper partially reporting how the reference standard was set for a local dataset through the use of a 'combination of detailed clinical history, clinical examination, oph-thalmic imaging, and/or neurological imaging' [39]. One further study employed an ambiguous description of ground truth where the presence or absence of a condition was annotated by 'one or multiple human readers for each image' where a combination of 'structured information' and 'manually-extracted textual information' was used [31]. Several studies used more than one ophthalmologist for annotation of a single image, thereby aiming to account for intra-rater variability [30,32,47,51,53].

**3.1.5 Discussion of diversity and patient demographics.** Seven studies used multi-ethnic datasets for training, though only two studies published in the last year provided a complete statistical summary of the ethnic groups in their dataset [32,39,41,44,53,54,62]. Through the use of heat maps, Milea et al. (2020) discussed the diversity in their dataset in terms of age, gender, country of origin, and ethnicity. They demonstrated instances in which the AI model perceived a 'pathological area' based on ethnicity.

## 3.2 Management and handling of data

**3.2.1 Storage of data.** Datasets used by studies that are publicly available such as STARE, DRIONS, ACRIMA, RFMiD, RFMiD 2.0, and Kim Ungsoo's Kaggle can be found online. Of those studies not using publicly available datasets,

Milea et al.'s (2020) BONSAI dataset is stored in the Singapore Eye Research Institute, and Avramidis et al.'s (2020) paediatric fundus dataset is stored in a Health Insurance Portability and Accountability Act-compliant Research Electronic Data Capture Database at University of Southern California. The remainder of the studies did not report the location of their data storage. STARE was the most commonly used online dataset source of papilloedema and normal retinal images by four studies [34–36,45]. Locally available datasets were used by 6 studies although details of data location were not reported [34–37,45,46].

### 3.2.2 Pre-processing steps in deep learning systems.

Pre-processing includes any steps that transform raw data through cleaning, noise reduction, removal of missing or inconsistent data, clustering, feature engineering, filtering, and modelling for improved and more efficient performance of the model [63]. Though a majority of the studies (n = 13) reported that they used pre-processing techniques, very few provided detailed information on the steps involved [30–33,38,40,43,47–50,53,55]. Standardisation of size and definition of region of interest at the optic disc were the two most common reasons for pre-processing fundus images [31 33, 40, 47-50, 53, 62]. Ahn et al. (2019) also adjusted variations in lighting and brightness through Gaussian filtering prior to cropping the image with the region of interest [33].

Quellec et al. (2020) performed illumination variations using a YCrCb colour space and converted all images to an RGB image for interpretation by the model [31]. Milea et al. (2020) developed an automated segmentation algorithm that performed a quality check, analysed the optic disk size, shape, margins, colour balance, exposure, brightness, and sharpness [32]. They performed automatic cropping of the optic disc image before classification of papilloedema occurred [32]. They also used U-Net which is another neural network framework for a pixel-level image segmentation to localise the optic disk [32]. Avramidis et al. (2022) developed an unsupervised optic disc detection algorithm based on intensity thresholding using blood vessels in that region and morphological operations [48].

### 3.2.3 Pre-processing steps in non-deep learning systems.

Optic disc detection and vessel segmentation to extract the region of interest was the most commonly used pre-processing step [34–36,45,52]. Naing and Aimmanee automate image segmentation by using factorized gradient vector flow, a special gradient vector flow for texture segmentation in oedematous optic discs, to localise the optic disc boundary with high accuracy [52]. Fatima et al. (2017) and Akbar et al (2017) also used a 2D Gabor wavelet for enhancement of blood vessels. Images were then cropped out and resized to standardise size and region of interest [35–38]. Akbar et al. (2017) also transformed their images into greyscale [34,35]. Echegeray et al. (2011) used an ophthalmic medical technician for cropping and resizing of digital photos and converted it into a right eye orientation as part of their standardisation process [37]. Johnson et al. (2018) uniquely used U-Net to create a blood vessel probability map for vessel inpainting as part of their pre-processing step [42]. Hagen et al. (2023) used a deep learning algorithm for extraction of the visible optic disc [46]. They also used a deep learning algorithm and image software program for semi-automatic segmentation of peripapillary arterioles and paired venules [46].

### 3.2.4 Handling of missing data, outliers, or data inconsistencies.

Six studies described how missing data, duplicates, or poor-quality data were managed [29,31–33,40,54]. Visual inspection and manual removal of poor-quality images, missing data, or mislabelled data was the most common method by all five studies. Additionally, Milea et al. (2020) also used a quality control algorithm and quoted a labelling error of 0.9% in their testing dataset [32].

### 3.2.5 Data augmentation techniques.

Data augmentation techniques are used to increase the size and quality of the training dataset to reduce overfitting and increase the overall generalisability of a model and are particularly valuable in the case of limited datasets [64]. Geometric transformations, random erasing, mixing images, feature space augmentations, and data warping are all methods for data augmentation of image datasets. The most commonly used methods to augment data in the included DLS studies were rotation and flipping [32,33,38,43,47,49–51,55]. Random dropout, adjustments to brightness and contrast, and cropping certain areas of the image were also used [32,33,48,49]. Images were typically reported to have been increased by 3, 5 or 10 times the original sample size by studies that reported these numbers [33,40,48].

**3.2.6 Handling of class imbalance.** Class imbalance refers to the issue of having a higher number of data points for one class such as normal retinal images compared to another class such as papilloedema, particularly in the training set [65]. Class imbalance can significantly reduce the overall performance and generalisability of model, particularly in unseen datasets. There are data level methods (oversampling or undersampling), algorithmic level methods (thresholding, cost sensitive learning, or one-class classification), and hybrid methods that have been proposed to overcome the issue of class imbalance [65]. Very few studies contained detailed reporting of how class imbalance was handled. Among those who did, Milea et al. (2020) performed data augmentation and set weights to their loss function to reduce any class imbalance [32] while Thiagarajan and Suguna (2023) trimmed the data to have a near equal number of images in each class [38]. Quellec et al. (2020) used a 'balanced dataset' such that all frequent conditions were equally represented, but did not apply this to rare conditions such as papilloedema [31].

## 3.3 Algorithm and model performance

**3.3.1 Study and participant characteristics.** We provide the summary statistics of all reported data in Table 3; age and gender of participants, number of images of normal optic disc and papilloedema in training and testing, and number of participants in training and testing.

Reporting rates for each parameter were inconsistent; for example, while 14 and 11 of the 19 DLS studies reported the number of images of papilloedema in their training and testing sets respectively, only 3 of 19 reported the number of participants in their study (for both training and testing sets), limiting the ability to provide a comprehensive overview of the study characteristics. There was significant variability in the number of images used to train and test the DLS models. The median number of images of normal optic discs used in the training set was 2509 (IQR 580–9156) and in the testing set was 569 (IQR 119–1378). The number of papilloedema images in the training and testing sets was lower with a median of 1292 (IQR 201–2882) in training set and 201 (IQR 57–388) in the testing set. This represents a significant degree of heterogeneity and class imbalance between the datasets. For the studies using non-DLS machine learning algorithms, insufficient numbers reported the above parameters to generate median and IQRs. Six models reported using multi-ethnic datasets, yet none provided a complete quantitative breakdown of the ethnic groups within their cohorts of participants. The reported representation of participants from high-income countries (HICs) was 2.4 times more in the included datasets when compared to participants from low-income countries (LMICs) (Fig 5).

**Table 3. Summary statistics of reported data in included studies.**

| | Study type | | | | | | | |
| --- | --- | --- | --- | --- | --- | --- | --- | --- |
| | Deep learning system (n = 19) | | Support vector machine (n = 6) | | Linear mixed model (n = 1) | | Random forest classifier (n = 1) | |
| | Studies reported | Median (IQR) | Studies reported | Median (IQR) | Studies reported | Pooled value | Studies reported | Median |
| Age (years) | 10 | 38.2 (29 – 48.6) | 0 | n/a | 1 | 42.3 | 0 | n/a |
| Gender (% female) | 10 | 62.7 (60.1 – 74.9) | 0 | n/a | 1 | 100 | 0 | n/a |
| Number of images of normal optic disc (training) | 11 | 2509 (580 – 9156) | 4 | 90 (50-245) | 1 | 0 | 1 | 0 |
| Number of images of papilloedema (training) | 14 | 1292 (201 - 2882) | 4 | 78 (70-93) | 1 | 0 | 1 | 0 |
| Number of images of normal optic disc (testing) | 10 | 569 (119 – 1378) | 0 | n/a | 1 | 5 | 1 | 88* (leave one out) |
| Number of images of papilloedema (testing) | 11 | 201 (57-388) | 0 | n/a | 1 | 18 | 1 | 88* |
| Number of participants (training) | 4 | 709.5 (279.5 – 3872) | 1 | 39 | 0 | NR | 1 | 88 |
| Number of participants (testing) | 3 | 111 (35-454) | 0 | n/a | 1 | 25 | 1 | 88 |

**Origin of participants in data set, categorised by HICs and LMICs**

Fig 5. Origin of participants in datasets as stated by authors in the included studies. The Organization for Economic Cooperation and Development (OCED) 2024/2025 categorization was used to defined HICs and LMICs. NR represents data that was not reported by studies.

**3.3.2  Model performance.**  A range of parameters as defined in S4 Appendix were used to measure the overall model performance, with accuracy, area under the receiving operator curve (AUROC), sensitivity, and specificity being the most commonly used metrics. Accuracy was the most commonly used performance metric by 14 DLS and 5 non-DLS studies, followed by sensitivity and specificity, which were reported by 12 DLS and 5 non-DLS studies. AUROC was used by 11 DLS and 2 non-DLS studies. F1 score and precision recall were reported by 4 DLS studies. PPV and NPV values were provided by one DLS and two non-DLS studies. The category 'others' include Cohen's weighted K coefficient provided by one non-DLS study, Spearman's rank provided by one DLS study, and precision value given by one DLS study.

Accuracy rates were more than 80% in 17 of the 19 studies which reported this value. Indicators of good performance such as high AUROC, accuracy, sensitivity, specificity or precision cannot be interpreted in isolation in studies using AI models. For instance, a model with one of the highest levels of reported accuracy (99.17%) was Saba et al. (2021)'s DLS which only included 60 images of normal retina and 40 images of papilloedema in their testing set, without any external validation. In contrast, Milea et al. (2020)'s robust reporting of their study provides a better overview of their BONSAI model's performance. BONSAI's reported accuracy was 94.8% for distinguishing papilloedema from other pathologies and normal retina in their internal validation dataset. This reduced to 87.5% when validated with an external dataset.

However, AUROC is generally preferred over accuracy as a measure of performance for binary classification, especially in imbalanced datasets [66]. AUROC measures the area under a Receiving Operator Characteristics (ROC) curve. It provides a measure of aggregated classification performance, allows comparison of performance between different classifiers, and helps examine trade-offs between true positive and false positive rates. Ranging from 0.5 to 1.0, 0.5 suggests that the test is no better than chance at making the distinction. Among four studies that performed external testing of their models [30,32,33,39], AUROC values between internal and external testing sets were reported to be very similar (see Table C in S5 Appendix) in three out of four of the studies that performed external testing, which reflects positively on the model's performance in being able to distinguish between normal retina and papiloedema images.

Of the 27 studies, ten provided the necessary data to construct contingency tables for a meta-analysis of diagnostic test accuracy using a random-effects model. Of these, seven studies used deep learning systems and three used non-DLS machine learning. Of these ten contingency tables, four were calculated from external test data, while the others did not specify if it was an external or internal test set. The sensitivity of the models among these studies ranged from

0.76-0.96 (mean = 0.89, SD = 0.07), while the specificity ranged from 0.69-0.98 (mean = 0.88, SD = 0.10). The total number of images in these testing sets ranged from 100-1886 (mean = 565, SD = 578). The $I^2$ value was calculated at 74%, indicating a large amount of heterogeneity between studies; however, it should be noted that in small meta-analyses the $I^2$ has been shown to be biased and should be interpreted with caution [67]. The pooled sensitivity and specificity were calculated to be 0.87 [95% CI 0.76-0.93] and 0.90 [95% CI 0.74-0.97], respectively (Fig 6).

These results show a strong diagnostic performance overall, with a high pooled sensitivity and specificity. However, the validity of this interpretation is limited by the small number of studies, a high degree of variability in the size of datasets, and high level of heterogeneity. Additionally, not all models were externally validated. It is also worth noting that only two of the four model types from this systematic review were included in the meta-analysis; neither of the studies using a linear mixed model or random forest classifier included the necessary data to be included in the meta-analysis.

### 3.3.3 Performance of DLS compared against clinicians.

Chang et al. developed an AI model to aid differentiating paediatric pseudopapilloedema from true papilloedema [54]. Their AI model was found to be most helpful in identifying mild papilloedema which can be challenging to diagnose in children. The model's sensitivity for diagnosing mild papilledeoma was significantly higher at 87.8% in external testing compared to the two human experts (53.1% and 49%) [53]. The performance of the BONSAI deep learning system was compared against both first line clinicians and expert neuro-ophthalmologists [28,32,41,51]. These studies included a relatively diverse cohort of 454 patients (800 images) where 60% were women with near equal proportions of Asian (48.7%) and Caucasian (50.9%) participants [28]. The mean age of participants was 44.5 (+/- 20.1) years [28]. The DLS outperformed 30 clinicians based in Singapore regardless of their ophthalmic training with an error rate of 15.3% [28]. By contrast, Emergency Department (ED) physicians had the highest error rate of 47.9% in detecting optic disc abnormalities including papilloedema [28]. The DLS was also found to be significantly faster (25 seconds) than 2 expert neuro-ophthalmologists (61 and 74 minutes) in classifying normal, papilloedema and other disc abnormality, though there was no statistically significant difference in terms of accuracy for papilloedema detection [41]. Accurate classification of the severity of the papilloedema (mild-moderate or severe) by the DLS (87.9%) was comparable to the ability of 3 neuro-ophthalmologists with an accuracy of 84.1%, 91.8%, and 73.9%, though these findings are not generalisable due to the very small sample cohort of neuro-ophthalmologists [44].

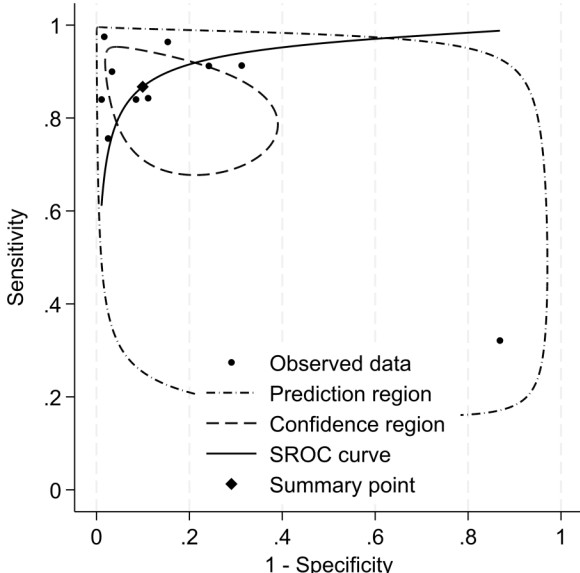

**Fig 6. Summary of receiving operator characteristics curve for seven studies included in the meta-analysis.**

### 3.3.4 Model interpretability.

In black box methods such as DLS models, visualisation using class activation maps can provide some insight into model's interpretation of abnormalities and help understand performance of the model, especially when images are misclassified. However, there are limitations to these methods for understanding decisions made by the model at an individual level and may not provide any 'performance guarantees' [68]. Only five deep learning studies used techniques to improve the interpretability of their model. Quellec et al. (2020) used heatmaps to show how much each pixel contributed to prediction of pathology, however not for papilloedema [31]. Cao et al. (2020) used feature maps to capture the regions on fundus photography and visual field tests where abnormality was detected. Milea et al. (2020),Branco et al. (2024), and Chang et al. (2024) used class activation mapping (CAM) to visualise the key features contributing to the neural networks' performance ( [32,53,54]). Milea et al. (2020) also found that there were differences in what the AI model perceived as 'pathological area' based on ethnicity [32]. Avramidis et al (2022) used saliency maps to confirm signs of papilloedema as interpreted by the model and verify the presence of these findings clinically [48]. Liu et al. (2023) performed CAM analyses for all the images used in their study and found that there were areas with 'strong activation' outside the optic disc in 32% of the eyes where the images were incorrectly classified [40]. Those that were correctly classified showed strong activation at the site of the disc.

### 3.4 Real-life clinical implications

A small proportion of studies (n = 3) discussed how their model may alleviate the workload of trained specialists, provide a low-cost approach, or be incorporated into telehealth monitoring, remote locations or screening programmes [38,39,62]. However, no study provided a description of its integration with existing workflows or demonstrated efficacy in improving care or clinical outcome.

### 3.5 Transparency, algorithmic bias, medicolegal and ethical implications

There was an absence of any detailed discussion surrounding algorithmic bias specifically introduced through their chosen methodology in the models designed for papilloedema detection. While some authors briefly acknowledged potential biases arising from inter-grader variability in annotating images, imbalanced datasets featuring a disproportionate number of papilloedema images, and lack of diverse datasets, a more thorough self-examination of models was not performed. Moreover, the model's potential to cause harm and any wider ethical, legal and social concerns of the model(s) were not addressed by any of the studies included in the review.

## 4. Discussion

Papilloedema is a clinical sign that can be associated with a wide range of potentially life-threatening medical conditions and can be challenging to identify by less experienced personnel. Advances in retinal fundus cameras which can be used in both non-mydriatic and mydriatic pupils have shown promise in ophthalmic telemedicine, however a major limitation is that the images are typically taken in specialised settings such as eye clinics, requiring skilled operators for image acquisition and specialists for interpretation [69]. The AI development pipeline for diagnosing papilloedema using retinal images may consist of several stages before it can be implemented widely (Fig 7). While AI can be used as a tool to analyse retinal images automatically, prospective clinical trials evaluating its diagnostic accuracy, cost-effectiveness, and comparison with health-care professionals and routine standard of care are required before it can be deployed safely. 'Human in the loop' approaches can ensure that healthcare professionals remain actively involved in reviewing, validating, and providing feedback about AI performance, and potentially improve the safety of deploying AI in real-world healthcare settings. Models must also be approved by regulatory bodies such as the FDA in the US or acquire the Conformité Européenne mark in Europe, or abide by the regulatory guidance developed by the Medicine and Health Regulatory Agencies (MHRA) on the use of 'software and AI' as a medical device in the UK.

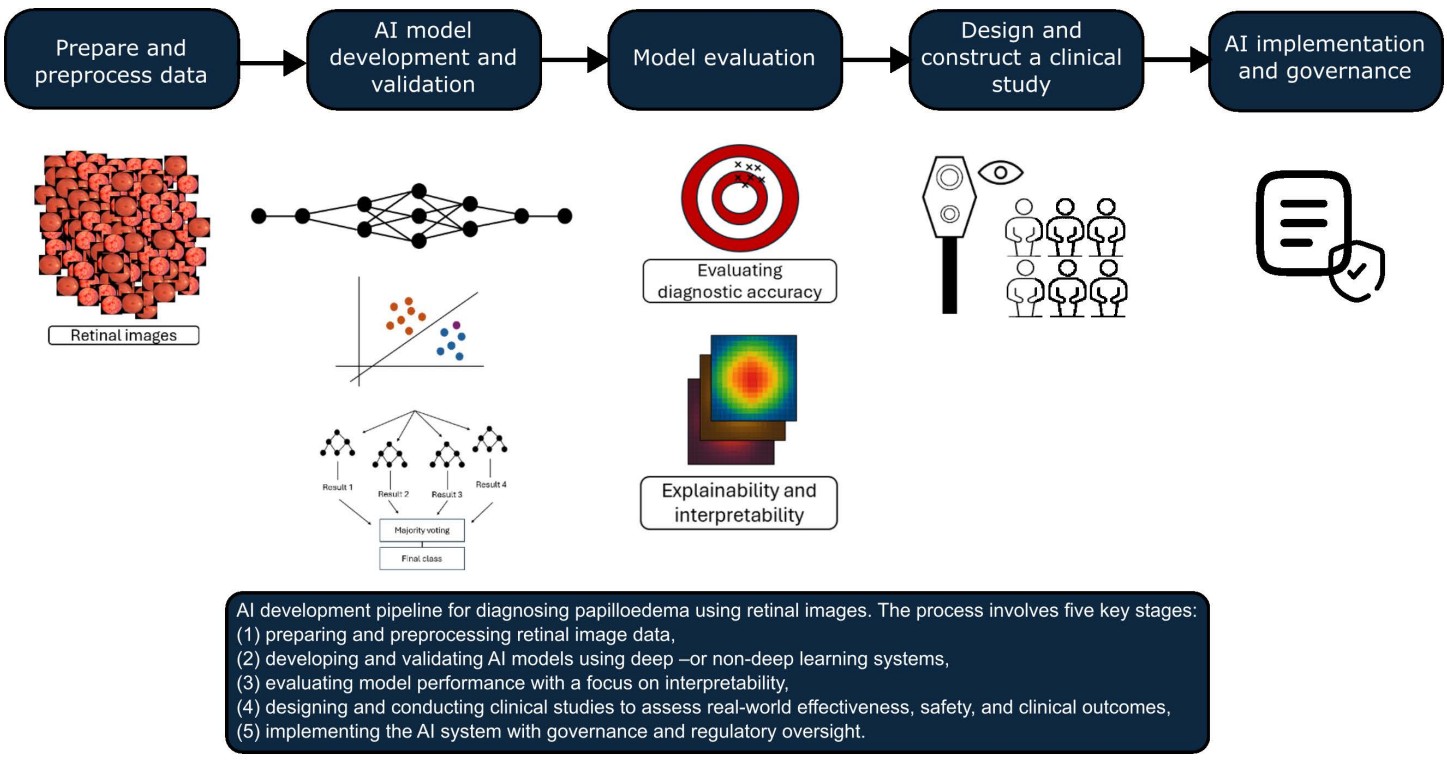

AI development pipeline for diagnosing papilloedema using retinal images. The process involves five key stages:
(1) preparing and preprocessing retinal image data,
(2) developing and validating AI models using deep –or non-deep learning systems,
(3) evaluating model performance with a focus on interpretability,
(4) designing and conducting clinical studies to assess real-world effectiveness, safety, and clinical outcomes,
(5) implementing the AI system with governance and regulatory oversight.

**Fig 7. AI development pipeline for diagnosing papilloedema using retinal images.**

The BONSAI-DLS stands out for its robust methodology, diverse multi-ethnic datasets, high levels of performance during external validation, and comparison with various healthcare professionals. However, despite these strengths, the study has limitations that prevent its use in a variety of clinical settings, notably the inability to distinguish between co-existing ophthalmological conditions [32]. Furthermore, the photographs used for development of the model were taken using expensive cameras, with diagnosis made by experts, which may pose a limitation in terms of transferability to low-resource settings. Interpretability was assessed using class activation maps, but the inherent nature of deep learning models means that they will inevitably possess some 'black box' characteristics.

In contrast, the non-DLS machine learning models identified provide the advantage of being able to extract specific features for training and testing, which may increase interpretability of AI models. However, the limiting factors are the risk of introducing bias in feature selection and challenges in detecting milder versions of papilloedema as shown by Fatima et al.'s (2017) study where the AFIO dataset with milder cases of papilloedema had lower accuracy dates for detection compared to STARE which contained mostly image of severe papilloedema [36].

A solution for maximising the strengths of both types of AI (non-DLS and DLS) while mitigating their limitations may mean using a mixture of methods such as in Hagen et al.'s (2023) study where the authors used a deep learning algorithm for the extraction of optic discs images and then employed a linear mixed model for analysis of arteriole-to-venule-ratios with corresponding lumbar opening pressures to estimate CSF pressure [46]. Though they collected data prospectively and set a reference standard that was objective for measuring ICP, the sample size of their study as well as their use of median rather than synchronised continuous ICP measurements (with arteriole-venule diameter ratio in the retina for correlation) may limit generalisability [46].

Automated papilloedema detection in resource-limited settings can potentially serve as a non-invasive, accessible diagnostic aide and a prognostic tool following brain injury. However, our review suggests that the existing state of the knowledge has several significant deficiencies. Chief among these is the lack of transparent, publicly available datasets from a large a varied global population with validated states of health and disease on which learning models can be trained. The paradox at the heart of this issue is that underserved populations in resource poor countries are exactly those same populations whose data is likely to be missing from large internationally available datasets, due to issues of healthcare access, technology, and skilled staff. If this is to be addressed, fundamental challenges must first be overcome: the generation of high-quality fundus images without the need for pupillary dilatation and expert ophthalmoscopy; the secure transmission and storage of these images into large interrogatable datasets; and standardised tools for setting ground truth with expert annotation confirming states of health and disease. Furthermore, applying appropriate processing techniques during the design and development of AI models can improve image quality, standardisation, algorithmic fairness, annotation accuracy, and help us implement quality control measures – ultimately improving model performance across data from varying levels of resources. For example, 'edge detection' as a preprocessing technique, can filter out insignificant information on an image and lead to better performance, but selecting the right type of edge detection algorithm is important due to trade-offs in computational cost and model performance [70]. Generative Adversarial Networks can also be used to enhance resolution of images and learn deep representations without extensively annotated training data, but it requires significant care in development due to risk of hallucinating features, difficulty in evaluating, and added computational cost [71].

Our systematic review and meta-analysis provides an overview of the existing literature on the detection of papilloedema using AI models. The studies included show high levels of performance for papilloedema detection. However, in the majority of the cases, the higher levels of performance were mainly limited to datasets that they were trained or tested on. Although we measured statistical heterogeneity in a subset of studies included in the meta-analysis, variation in reporting quality, limited access to underlying datasets, and variability in performance metrics chosen by authors affect the choice of meta-analytic parameters and statistical tests, limiting our ability to draw more conclusive insights. Biases in study selection included in the meta-analysis which may be affected by publication bias and variation in reporting styles between journals, can also affect pooled diagnostic accuracy. We also identified several major limitations that prevent clinical translation of majority of the models such as poor quality of reporting, lack of external validation, lack of diverse datasets for training and testing, ambiguous definitions for ground truth, lack of interpretability in black box methods, retrospective data collection, and lack of comparison of performance with healthcare professionals. The lack of reporting of demographic data such as gender, ethnicity, country of origin, and socioeconomic status can risk exacerbating health inequalities by training models on biased datasets and allowing further biases to perpetuate during its development. More robust methodologies for evaluating risk of bias, model safety, and model performance are required before they can be used in either high or low resource settings.

## Supporting information

**S1 Appendix. Search strategy and definitions.**
(DOCX)

**S2 Appendix. PCC Criteria for study selection and the PRISMA checklist.**
(DOCX)

**S3 Appendix. Assessment of reporting quality using checklist for artificial intelligence in medical imaging.**
(DOCX)

**S4 Appendix. Commonly used metrics for measuring performance and their definition.**
(DOCX)

**S5 Appendix. Supplementary data tables.**
(DOCX)

**S6 Appendix. Table of articles screened and reasons for exclusion.**
(DOCX)

**S1 File. This file which can be found in S6 Appendix contains a detailed table with all the articles screened and reasons for exclusion.**
(XLSX)

## Acknowledgments

The members of the EyeVu Consortium include Abdulhakeem Abubakar Tunde, Abdur Raafay Iqbal, Alex Lawrence, Andrea Cuschieri, Antonia Vogt, Anyela Flor Bruno Peña, Ayda Lazemi, Blendi Bylygbashi, Charles Britton, Chiara Spezzani, Christos Antonopoulos, Daniel Black Boada, Daniel Shao, Dipanshu Gandhi, Ekwegbara Somtochukwu Mitchel, Elena Maerz Engstler, Emmanuel Chileshe Phiri, Geneviève Endalle, Ghina Hussain, Kassim Omar Kassim, Kehinde Alare, Kübra Tamer, Leona Takeuchi, Makinah Haq, Marwa SaedAli Emhemed, Mubarak Mustapha Jolayemi, Muhammad Iqbal Aniq, Nagheli Fernanda Borjas-Calderón, Ngepgou Beckline Tazoah, Nneka Lilian Amakom, O. Joshua Sokan, Olaoluwa Ezekiel Dada, Olobatoke Tunde Ayomide, Oloruntoba Ogunfolaji, Phupha Amornkijja, Razan Eid, Roshen Sidhu, Rushi Patel, Shodip Shrestha, Sruthi Ranganathan, Tangmi Djabo Eric Adrien, Temitayo Ayantayo, Tom Wilkins, Weng Tong Wu, Wesley Barrett, Zafer Utku Ulker.

## Author contributions

**Conceptualization:** Lekaashree Rambabu.

**Formal analysis:** Lekaashree Rambabu, Thomas Edmiston.

**Investigation:** Lekaashree Rambabu, Thomas Edmiston, Brandon Smith.

**Methodology:** Lekaashree Rambabu, Thomas Edmiston.

**Project administration:** Lekaashree Rambabu.

**Supervision:** Peter J. Hutchinson, Tom Bashford.

**Writing – original draft:** Lekaashree Rambabu.

**Writing – review & editing:** Lekaashree Rambabu, Thomas Edmiston, Brandon G. Smith, Katharina Kohler, Angelos G. Kolias, Richard A.I. Bethlehem, Pearse A Keane, Hani Marcus, Peter J. Hutchinson, Tom Bashford.

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
