## [Decision Letter · Decision Letter 0]

8 Apr 2025

PDIG-D-25-00090Detecting papilloedema as a marker of raised intracranial pressure using artificial intelligence: a systematic reviewPLOS Digital Health Dear Dr. Rambabu, Thank you for submitting your manuscript to PLOS Digital Health. After careful consideration, we feel that it has merit but does not fully meet PLOS Digital Health's publication criteria as it currently stands. Therefore, we invite you to submit a revised version of the manuscript that addresses the points raised during the review process. Please submit your revised manuscript within 60 days Jun 07 2025 11:59PM. If you will need more time than this to complete your revisions, please reply to this message or contact the journal office at digitalhealth@plos.org. Please include the following items when submitting your revised manuscript:* A rebuttal letter that responds to each point raised by the editor and reviewer(s). You should upload this letter as a separate file labeled 'Response to Reviewers '. This file does not need to include responses to any formatting updates and technical items listed in the 'Journal Requirements' section below.* A marked-up copy of your manuscript that highlights changes made to the original version. You should upload this as a separate file labeled 'Revised Manuscript with Track Changes '.* An unmarked version of your revised paper without tracked changes. You should upload this as a separate file labeled 'Manuscript '. If you would like to make changes to your financial disclosure, competing interests statement, or data availability statement, please make these updates within the submission form at the time of resubmission. Guidelines for resubmitting your figure files are available below the reviewer comments at the end of this letter. We look forward to receiving your revised manuscript. Kind regards, Gloria Hyunjung KwakSection EditorPLOS Digital Health Gloria Hyunjung KwakSection EditorPLOS Digital Health Leo Anthony CeliEditor-in-ChiefPLOS Digital Healthorcid.org/0000-0001-6712-6626 **Journal Requirements:**1. Please send a completed 'Competing Interests' statement, including any COIs declared by your co-authors. If you have no competing interests to declare, please state "The authors have declared that no competing interests exist". Otherwise please declare all competing interests beginning with the statement "I have read the journal's policy and the authors of this manuscript have the following competing interests:" 2. Please provide separate figure files in .tif or .eps format. For more information about figure files please see our guidelines:  https://journals.plos.org/digitalhealth/s/figures https://journals.plos.org/digitalhealth/s/figures#loc-file-requirements 3. We have noticed that you have uploaded Supporting Information files, but you have not included a list of legends. Please add a full list of legends for your Supporting Information files after the references list. 4. As required by our policy on Data Availability, please ensure your manuscript or supplementary information includes the following:  A numbered table of all studies identified in the literature search, including those that were excluded from the analyses.   For every excluded study, the table should list the reason(s) for exclusion.   If any of the included studies are unpublished, include a link (URL) to the primary source or detailed information about how the content can be accessed.  A table of all data extracted from the primary research sources for the systematic review and/or meta-analysis. The table must include the following information for each study:  Name of data extractors and date of data extraction  Confirmation that the study was eligible to be included in the review.   All data extracted from each study for the reported systematic review and/or meta-analysis that would be needed to replicate your analyses.  If data or supporting information were obtained from another source (e.g. correspondence with the author of the original research article), please provide the source of data and dates on which the data/information were obtained by your research group.  If applicable for your analysis, a table showing the completed risk of bias and quality/certainty assessments for each study or outcome.  Please ensure this is provided for each domain or parameter assessed. For example, if you used the Cochrane risk-of-bias tool for randomized trials, provide answers to each of the signalling questions for each study. If you used GRADE to assess certainty of evidence, provide judgements about each of the quality of evidence factor. This should be provided for each outcome.   An explanation of how missing data were handled.  This information can be included in the main text, supplementary information, or relevant data repository. Please note that providing these underlying data is a requirement for publication in this journal, and if these data are not provided your manuscript might be rejected.   5. Please provide a complete Data Availability Statement in the submission form, ensuring you include all necessary access information or a reason for why you are unable to make your data freely accessible. If your research concerns only data provided within your submission, please write "All data are in the manuscript and/or supporting information files" as your Data Availability Statement.**Additional Editor Comments (if provided):****Reviewers' Comments:** Reviewer's Responses to Questions

**Comments to the Author**

1. Does this manuscript meet PLOS Digital Health’s publication criteria ? Is the manuscript technically sound, and do the data support the conclusions? The manuscript must describe methodologically and ethically rigorous research with conclusions that are appropriately drawn based on the data presented.

Reviewer #1: Yes

Reviewer #2: Yes

Reviewer #3: Yes

2. Has the statistical analysis been performed appropriately and rigorously?

Reviewer #1: Yes

Reviewer #2: Yes

Reviewer #3: Yes

3. Have the authors made all data underlying the findings in their manuscript fully available (please refer to the Data Availability Statement at the start of the manuscript PDF file)?

Reviewer #1: Yes

Reviewer #2: Yes

Reviewer #3: Yes

4. Is the manuscript presented in an intelligible fashion and written in standard English?

Reviewer #1: Yes

Reviewer #2: Yes

Reviewer #3: Yes

5. Review Comments to the Author

Reviewer #1: This systematic review evaluated the use of AI for detecting papilloedema from retinal images, analyzing 27 systems (19 deep learning, 8 non-deep learning). Studies showed high variability in training/testing image numbers, with limited demographic data and only 10 studies performing external validation. Pooled sensitivity and specificity were 0.87 and 0.90, respectively. Despite promising performance, data bias, poor reporting, and lack of reproducibility limit confidence. AI models could support papilloedema triage in resource-limited settings, but robust external validation across diverse populations and clinical environments is essential before routine clinical adoption can be recommended. This review is interesting and valuable, however there is a few concerns to publish in current version.

Major concerns

1. The authors correctly highlight that many studies fail to meet the CLAIM criteria. However, the underlying issue is that a significant proportion of studies were not originally designed to conform to these criteria. In other words, the authors are raising the important question of whether CLAIM itself represents the optimal standard for evaluating these studies. I agree that the approach of seeking the most appropriate evaluation framework, rather than rigidly adhering to one specific set of criteria, is appropriate.

2. The authors also emphasize the importance of generating high-quality fundus images without the need for pupil dilation or specialist optometry, ensuring the safe transmission and storage of these images in large, collatable datasets, and using expert annotations to assess health and disease states and to develop benchmark tools. These are all critical points. However, given current practical and logistical constraints, it may be unrealistic to expect widespread adoption of these processes in the near future. It would be valuable if the authors could propose strategies for leveraging the insights gained from the substantial body of existing research and explore potential methods for integrating and harmonizing this body of work with future studies.

3. The authors highlight image quality as a recurring issue — a point that has been previously noted in the literature. To strengthen this argument, it would be beneficial to explicitly reference prior studies that have addressed the challenges of image standardization across different manufacturers and imaging platforms. Addressing this industry-wide inconsistency with supporting citations would help contextualize the problem and underscore the need for standardized imaging protocols.

Reviewer #2: This systematic review addresses an important clinical problem: the automated detection of papilloedema an ocular manifestation of raised intracranial pressure using artificial intelligence (AI) applied to retinal images. Given the potential impact on triage in resource‐limited settings, the review is timely and relevant.You comprehensively searched multiple databases and included both deep learning and traditional machine learning studies. You assessed the quality of reporting using established checklists (CLAIM) and introduced a novel rubric (SMART) to appraise bias and methodological robustness. Furthermore, the manuscript presents summary statistics and meta-analytic estimates of diagnostic performance. Overall, the work is ambitious in scope and offers valuable insights into the current state of AI applications in papilloedema detection.

Below are some strengths.

Major Strengths

1. Clinical Relevance and Innovation:

The review targets a crucial unmet need—rapid and accurate identification of raised intracranial pressure via papilloedema detection, which could be transformative in low-resource settings. The integration of AI into bedside ophthalmic screening is both innovative and of high clinical importance.

2. Methodological Rigour:

The authors have followed systematic review standards (PRISMA 2020) and prospectively registered the protocol with PROSPERO. Their search strategy spans four major databases, ensuring a wide capture of the relevant literature.

3. Dual Assessment of Reporting Quality and Bias:

The use of the CLAIM checklist for assessing reporting standards is commendable. Moreover, introducing the SMART rubric to identify potential biases in AI studies adds depth to the evaluation and highlights key areas needing improvement in the primary literature.

4. Quantitative Synthesis:

The meta-analysis of sensitivity and specificity provides an aggregated measure of diagnostic performance. Reporting pooled estimates helps readers understand the overall accuracy of AI systems, even though heterogeneity remains a challenge.

5. Comprehensive Data Extraction:

The manuscript offers detailed tables summarizing study characteristics, dataset attributes, and model performance metrics. Such granularity is useful for readers seeking to understand the landscape of methodologies and performance variations.

Also here are some weaknesses I think should be addressed

Major Weaknesses

1. Heterogeneity and Limited External Validation:

A recurring issue highlighted by the review is the significant heterogeneity among studies—ranging from dataset size to image quality and annotation standards. Only a fraction of studies performed external validation, which raises concerns about the generalizability of the reported performance metrics. The review would benefit from a more in-depth discussion of how heterogeneity impacts the meta-analytic estimates and the potential for publication bias.

2. Quality of Reporting in Primary Studies:

The review notes that many primary studies do not fully adhere to reporting standards, especially in the results sections where clinical and demographic data are often missing. This limits the ability to critically evaluate the validity of the AI models. The manuscript could further explore how this poor reporting may bias the meta-analysis results.

3. Inclusion and Exclusion Criteria:

Although the review details the screening process, it is not entirely clear whether the inclusion criteria for studies (e.g., types of retinal images, definitions of papilloedema) were sufficiently stringent. Greater clarity regarding the PCC criteria would strengthen the review’s reproducibility.

4. Novel SMART Rubric:

While the SMART rubric is an innovative tool for assessing bias in AI studies, its development and validation are not described in sufficient detail. More information on its derivation, weighting, and inter-rater reliability would increase confidence in its use and findings.

5. Discussion of Clinical Implementation:

Although the review acknowledges that AI models show promise, there is limited discussion on the practical challenges of translating these findings into clinical practice. Topics such as regulatory approval, integration with existing ophthalmic devices, and the need for prospective validation in diverse populations deserve further exploration.

Here are some Suggestions for Improvement

1. Clarify Heterogeneity and Its Impact:

Expand the discussion on how variability in training/testing dataset sizes, image quality, and annotation methods might affect pooled sensitivity and specificity. Consider performing subgroup analyses (if data permit) to determine whether certain study characteristics correlate with better performance metrics.

2. Enhance Reporting on Bias Assessment:

Provide a more detailed description of the SMART rubric, including its domains, scoring system, and validation process. Discuss how the rubric’s findings compare with those from other established bias tools in AI research.

3. Address External Validity:

Emphasise the need for prospective studies with external validation. Consider adding recommendations for future research, such as multi-center collaborations and the use of standardized imaging protocols, to improve the generalizability of AI algorithms.

4. Improve the Discussion Section:

The discussion would benefit from a deeper exploration of the clinical implications of the findings. How might the integration of these AI systems impact current diagnostic workflows? What are the potential cost-benefit and ethical considerations when implementing AI in low-resource settings?

5. Limitations of the Meta-analysis:

Acknowledge the limitations inherent in aggregating data from studies with diverse methodologies and reporting standards. Discuss potential biases in study selection (e.g., publication bias) and how they might affect the pooled diagnostic accuracy.

6. Future Research Directions:

Outline concrete recommendations for improving study design in future research. This might include advocating for larger, diverse datasets, clearer definitions of papilloedema, standardized reporting guidelines, and the inclusion of clinical outcomes beyond imaging metrics.

Minor Comments

- Terminology Consistency:

Ensure that terms such as “papilloedema,” “papilledema,” and “optic disc swelling” are used consistently throughout the manuscript to avoid confusion.

- Formatting and Tables:

Some tables (e.g., those summarizing dataset characteristics and performance metrics) are very dense. Consider simplifying or summarizing key points in the text, with detailed tables provided in supplementary materials.

- Reference Standard Discussion:

The manuscript could benefit from a clearer explanation of the various “ground truth” methods used across studies. A comparative analysis of these methods would help elucidate their impact on reported performance.

- Ethnicity and Demographics:

Given the observed under-reporting of participant demographics, particularly ethnicity, the authors should stress the importance of collecting and reporting these data to understand model performance

Conclusion

This systematic review represents an important contribution to the field of AI in ophthalmology, specifically for the detection of papilloedema as a marker for raised intracranial pressure. The comprehensive search strategy, dual approach to quality and bias assessment, and quantitative synthesis of diagnostic performance are notable strengths. However, the manuscript also highlights significant challenges, including heterogeneity among studies, limited external validation, and suboptimal reporting standards. Addressing these issues in future research is critical for the translation of AI models into clinical practice.

Overall, with revisions to address the concerns noted above—particularly a deeper discussion on heterogeneity, external validity, and the clinical implementation of AI systems—the manuscript has the potential to serve as a valuable resource for both clinicians and researchers working at the intersection of neuro-ophthalmology and artificial intelligence.

I hope these comments are helpful in strengthening your manuscript.

Reviewer #3: This systematic review addresses a highly relevant and clinically significant topic—papilledema detection as an importante marker for elevated intracranial pressure. Given the potentially life-threatening nature of conditions associated with papilledema, timely and accurate identification is essential for appropriate management.

The authors provide a comprehensive and thoughtful overview of the current state of artificial intelligence applications in this area. The review thoroughly covers the imaging acquisition process, characteristics of the datasets and algorithms used, as well as technical aspects such as performance metrics. Notably, the authors also acknowledge the limitations and potential sources of bias in the existing literature, which strengthens the critical rigor of the manuscript.

To enhance the accessibility and clarity of the review for readers from non-specialist backgrounds (e.g., general practitioners, emergency physicians, or AI researchers), I suggest including illustrative figures with representative examples of papilledema and normal optic discs. Visual aids would help contextualize the clinical relevance of the findings and improve comprehension, especially for audiences without ophthalmology or neurology training.

Overall, this is a valuable contribution to the field of AI in neuro-ophthalmology.

6. PLOS authors have the option to publish the peer review history of their article (what does this mean? ). If published, this will include your full peer review and any attached files.

**Do you want your identity to be public for this peer review?** For information about this choice, including consent withdrawal, please see our Privacy Policy .

Reviewer #1: **Yes: ** Toru Miwa

Reviewer #2: No

Reviewer #3: No

---

## [Decision Letter · Decision Letter 1]

23 Jul 2025

Detecting papilloedema as a marker of raised intracranial pressure using artificial intelligence: a systematic review

PDIG-D-25-00090R1

Dear Dr Rambabu,

We are pleased to inform you that your manuscript 'Detecting papilloedema as a marker of raised intracranial pressure using artificial intelligence: a systematic review' has been provisionally accepted for publication in PLOS Digital Health.

Best regards,

Gloria Hyunjung Kwak

Section Editor

PLOS Digital Health

**Additional Editor Comments (if provided):**

**Reviewer Comments (if any, and for reference):**

Reviewer's Responses to Questions

**Comments to the Author**

1. If the authors have adequately addressed your comments raised in a previous round of review and you feel that this manuscript is now acceptable for publication, you may indicate that here to bypass the “Comments to the Author” section, enter your conflict of interest statement in the “Confidential to Editor” section, and submit your "Accept" recommendation.

Reviewer #1: All comments have been addressed

2. Does this manuscript meet PLOS Digital Health’s publication criteria ? Is the manuscript technically sound, and do the data support the conclusions? The manuscript must describe methodologically and ethically rigorous research with conclusions that are appropriately drawn based on the data presented.

Reviewer #1: Yes

3. Has the statistical analysis been performed appropriately and rigorously?

Reviewer #1: Yes

4. Have the authors made all data underlying the findings in their manuscript fully available (please refer to the Data Availability Statement at the start of the manuscript PDF file)?

Reviewer #1: Yes

5. Is the manuscript presented in an intelligible fashion and written in standard English?

Reviewer #1: Yes

6. Review Comments to the Author

Reviewer #1: All concerns are appropriately corrected.

7. PLOS authors have the option to publish the peer review history of their article (what does this mean? ). If published, this will include your full peer review and any attached files.

**Do you want your identity to be public for this peer review?** For information about this choice, including consent withdrawal, please see our Privacy Policy .

Reviewer #1: **Yes: ** Toru Miwa
